# AI-Based Multi Sensor Fusion for Smart Decision Making: A Bi-Functional System for Single Sensor Evaluation in a Classification Task

**DOI:** 10.3390/s21134405

**Published:** 2021-06-27

**Authors:** Feryel Zoghlami, Marika Kaden, Thomas Villmann, Germar Schneider, Harald Heinrich

**Affiliations:** 1Automation, Maintenance and Factory Integration, Infineon Technologies Dresden GmbH & Co. KG, 01099 Dresden, Germany; germar.schneider@infineon.com (G.S.); harald.heinrich@infineon.com (H.H.); 2Computational Intelligence, University of Applied Sciences Mittweida, 09648 Mittweda, Germany; kaden1@hs-mittweida.de (M.K.); thomas.villmann@hs-mittweida.de (T.V.)

**Keywords:** sensor fusion, sensor evaluation, prototype-based learning, classification, artificial intelligence

## Abstract

Sensor fusion has gained a great deal of attention in recent years. It is used as an application tool in many different fields, especially the semiconductor, automotive, and medical industries. However, this field of research, regardless of the field of application, still presents different challenges concerning the choice of the sensors to be combined and the fusion architecture to be developed. To decrease application costs and engineering efforts, it is very important to analyze the sensors’ data beforehand once the application target is defined. This pre-analysis is a basic step to establish a working environment with fewer misclassification cases and high safety. One promising approach to do so is to analyze the system using deep neural networks. The disadvantages of this approach are mainly the required huge storage capacity, the big training effort, and that these networks are difficult to interpret. In this paper, we focus on developing a smart and interpretable bi-functional artificial intelligence (AI) system, which has to discriminate the combined data regarding predefined classes. Furthermore, the system can evaluate the single source signals used in the classification task. The evaluation here covers each sensor contribution and robustness. More precisely, we train a smart and interpretable prototype-based neural network, which learns automatically to weight the influence of the sensors for the classification decision. Moreover, the prototype-based classifier is equipped with a reject option to measure classification certainty. To validate our approach’s efficiency, we refer to different industrial sensor fusion applications.

## 1. Introduction

With the introduction of Industry 4.0 level [1], we count the integration of new sensor technologies in combination with AI-based solutions in real-world applications every day. The topic is called smart sensors [2], which participate in the improvement of productivity and the increase of the turnovers of many industries. These benefits have been confirmed, especially when there is an efficient application of the technologies offered in the market. However, AI applications can be dangerous in some cases and run the concerned company into serious problems. Besides, sensors can behave differently from one environment to another. They may deliver data with different qualities [3], which in some cases confuse the AI model and cause classification failures if the model is not stable enough. One misclassification case is costly, apart from the big effort and high expenses that concern the development of an AI-based system dedicated to solving one classification problem. Depending on the application field, the misclassification drawbacks vary from one domain to another. For example, on one hand in medicine, like that illustrated in [4], when the digital diagnostic reveals that a person is sick, while in reality, his health is not that bad. The doctor can later double-check and realize that the person is actually healthy. However, in the reverse case, not identifying a patient who is sick and then letting him go without any treatment is very dangerous. Here, human safety is considered, and therefore high classification accuracy is required, at values more than 99.99%. On the other hand, in most industrial applications, we tolerate slightly higher classification errors for the applications that do not menace people’s security. The paper of Xiong, Y., and Zuo, R. [5], as well as of Wossen, T et al. [6] based on different experiments in the agriculture field, confirmed that the cost of the misclassification error differs if it is a false negative error or false positive error in financial and material loss. To obtain low classification error rates, different defect prediction methods have been proposed in the literature [7,8,9]. Other methods dedicated to reducing misclassification cases based on support vector machines (SVMs) have been proposed in [10]. These methods rely on a large number of training data, i.e., data with class information. This phase remains difficult and time-consuming, especially when dealing with new software projects or existing projects without historical defect data. In addition, these methods focus on the post-evaluation of the classification process, and therefore time is an important factor here; reaching a minimum classification error rate with minimum possible risks is required. After estimation of the misclassification impact based on benchmark data, further model re-trainings should be performed, as illustrated by Xiong, Y., and Zuo, R. in their paper [5], where they explore the effect of misclassification errors to train a cost-sensitive neural network with different cost ratios. Although it obtained good results, this approach is time-consuming and needs an online training parameter update. Several other types of research exist, which are interested in an online evaluation of sensor data. Song, Y et al. in their paper [11] choose to refer to evidence theory and intuitionistic fuzzy in order to dynamically evaluate the reliability of sensors data. As a result, the system is able to assign a reasonable reliability factor to sensors that provide conflicting information. However, the proposed system architecture presents a high complexity and needs a strong mathematical background.

Fusing data from different sources for decision making and system control provides big benefits, mainly in terms of the higher confidence and better resolved system information. However, it is a big challenge to decide which sensors should be combined for a given task. Respective considerations are usually time consuming and costly. As an alternative, we propose a smart AI-based approach to intelligent sensors fusion to determine several sensor information contributions in dependence on the specific requirements, circumstances and tasks. In particular, we suggest applying an interpretable AI-model as pre-evaluation. This strategy of AI-based sensor evaluation and intelligent fusion by means of interpretable models can be easily adapted to many sensor fusion applications. In this way, the interpretability of the model enables the applicant to follow the decision making process. Furthermore, the proposed approach integrates the possibility to reject a query if the model is uncertain regarding a decision. We show how to build up a bi-functional system integrating both features. We focus on both the dynamic and static pre-evaluation of the considered system. A dynamic pre-evaluation refers to an evaluation of the sensor inputs during the training process of the classifiers, whereas the static pre-evaluation is performed offline once the classifiers are trained, but before using them in the production lines. In fact, the obtained system allows the evaluation of every single sensor regarding the provided data contribution in the predefined classification task and the robustness of these data under different external conditions. Our approach has the advantage of a parallel mechanism of both the classification and evaluation process. Besides, the evaluation system we propose in this study is adaptable and independent of the fusion system to be considered as well as of the classification purposes. We refer in the development of the evaluation system to the generalized learning vector quantization (GLVQ) [12], as a cost function-based extension from the LVQ [13,14], which provides the ability to track the data flow during the training time and makes the interpretation of the training model easier. In fact, the system learns, for each class, prototypes, which are responsible for the classification decision, as well as feature weighting vectors which present relevance for the discrimination of different classes [15]. Furthermore, we implement the reject option in the classification process [16,17] in order to increase the certainty of the interpretable results. These evaluation features, which are proposed in this paper, will allow engineers to decide on the best sensor combination to adapt in the fusion system and how far one sensor can be trusted to perform a certain classification task. This study is an extension of the evaluation tool presented in our recently published paper [15].

The paper is organized as follows. In Section 2, the proposed approach is explained in detail, while in Section 3, the application of the method and the validation of its efficiency are presented based on several examples. Finally, Section 4 concludes the study and opens the doors for further future work.

## 2. Methodology

The proposed approach focuses on sensor fusion systems that are dedicated to solve a classification problem. A general overview about the proposed system functionalities regarding the classification and evaluation parts, is presented in the Figure 1.

### 2.1. Prototype-Based Classification

In prototype-based classification, the LVQ is a learning method, which concerns small and not complicated classification problems with the advantage of being human interpretable. For problems with less complexity in the classification architecture and especially when we refer to a relatively small data set for training, using a deep neural network (DNN) would be overdone. From one hand, they may achieve very good results, but on the other hand, they remain time- and power-consuming. Besides, the data flow during training is difficult to track or to interpret. The standard LVQ itself is not particularly powerful when compared to DNNs, however, it has some extensions that can make the algorithm a powerful tool in a variety of classification-related tasks. In this paper, we refer to the generalized LVQ (GLVQ, [18]) based on a cost function approximating the classification error and implicitly optimizes the hypothesis margin [19]. In general, LVQ and its extensions consist in prototypes training, taking the distances to training data points $x_{k_{|k=1,\ldots ,N}}\in X\subset R^{n}$ with class labels $y\left(x_{k}\right)\in Y=\left\{y_{c}\in N|c=1,\ldots ,C\right\}$ into account. More precisely, each prototype $w_{i}\in W\subset R^{n}$ is additionally equipped with a class label $y(w_{i})\in Y$, such that at least one prototype per class is available. The GLVQ cost function [18] to be minimized is defined as:(1)$$\begin{array}{ccc} L(X,W) & = & \sum_{i=1}^{N}f\left(\mu \left(x_{i},W\right)\right) \end{array}$$
with a monotonously increasing function *f* like the sigmoid function. The so-called classifier function
(2)$$\begin{array}{ccc} \mu (x_{i},W) & = & \frac{d\left(x_{i},w^{+}\right)-d\left(x_{i},w^{-}\right)}{d\left(x_{i},w^{+}\right)+d\left(x_{i},w^{-}\right)} \end{array}$$
uses for each data point the distance to the best matching prototype $w^{+}$ with a consistent class label and to the best matching prototype $w^{+}$ of another class, i.e., $y\left(x_{k}\right)\neq y\left(w^{-}\right)$. The best matching prototype has a minimal distance compared to others with the same property. We apply for comparison the squared Euclidean distance here:(3)$$d\left(x_{k},w_{i}\right)=\left|\right|x_{k}-w_{i}{\left|\right|}_{E}^{2},$$
but any other dissimilarity like Manhattan distance [20], divergences [21], or kernel-based distances [22] can be used. In the recall phase, the winner-takes-all rule is applied. Thereby, the class of the best matching prototype is assigned to a new data point. The obtained classifier based on the trained prototypes will be used during the testing phase for new data points classification as well as for the sensors evaluation.

### 2.2. Sensors Evaluation in Fusion Systems

Since we are interested in sensor fusion systems, before running trainings using GLVQ, samples should be collected from each of the considered sensors. Our approach starts by creating small data sets for each sensor. Samples should be aligned in time and space. In fact, after installing the sensors in one compact system, if two or more sensors share the same information data, these data should be calibrated and brought to the same reference system. For example, for 3D sensors, the collected 3D position should be brought to one unique coordinate system that belongs to one of the 3D sensors or brought to one predefined virtual coordinate system. The calibration parameters should be stored and used later during the inference time. An external (physical or virtual via software) trigger connected to all sensors should be used in order to make sure of a simultaneous data recording from all sensors in the fusion system.

In this study, on the one hand, we focus on the evaluation of sensors regarding the importance and contribution of the samples in the predefined classification task [15]. On the other hand, the evaluation concerns the robustness of each sensor, with and without the presence of external noise. Our approach is based on both online and offline evaluation.

#### 2.2.1. Sensors Contribution in Classification Problems

This part is focusing on dynamic fusion systems. The system architecture in Figure 2 presents a bi-functionality workflow. Successive layers are presented with corresponding input dimensions.

As an output, the user obtains a prototype-based classifier, as well as the weight of the contribution for each sensor in the classification task designed by the parameters $\alpha_{s}$. The advantage of this approach is that both functionalities run in parallel. The dataflow is summarized in the following steps:(A)Data preparation: data sets creation for each of the considered sensors(B)Feature extraction: relevant patterns extraction out of the training data points used for prototypes training(C)Prototype-based classification: training parameters definition corresponding to the distance layer(D)Sensors contribution: Sensor’s data weighting depending on each sensor contribution in the classification task

In the following section, a detailed explanation of the cited steps is presented for a better understanding of the proposed approach.

##### Data Preparation

After calibration and assuring a synchronization of the data coming from different sensors in the fusion system like explained in the Section 2.1, for each sensor $S_{i}$, one data set $X_{i}=\left\{x_{j}^{i}|j=1,\ldots ,N\right\}$ is created, where *N* is the total number of samples that we collect from each sensor. Moreover, we define the set of classes $Y=\{y_{c}\in N|c=1,\ldots ,C\}$ where *C* is the number of classes. For each $x_{j}^{i}$, a unique $y_{c}$ is assigned, which is the same for each sensor.

##### Feature Extraction

Feature extraction corresponds to reduce the number of input variables into useful patterns when developing a predictive model. This step consists of building a neural network composed mainly of convolutional layers dedicated to extracting relevant features from the input data. Since the data that we use for training are labeled, we know in advance the target variables, we adapt a supervised feature selection method. The feature extraction operation is performed separately on the sensor input data. The output feature vectors $F_{i}=\left\{f_{j}^{i}\in R^{n^{i}}|j=1,\ldots ,F^{i}\right\}$ corresponding to the data set $X_{i}$ from a single data source $S_{i}$ are stored to be used for the prototype-based classification training.

##### Prototype-Based Classification

At this stage, the extracted features in the previous step serve as input to a new layer called distance layer [14]. This layer replaces the communally used fully connected layer also known as the dense layer responsible for the last classification decision. The distance layer is connected to the feature extraction network and it takes care of the prototype learning process. For each predefined class, one or more prototypes are initiated. One prototype is a distance-vector $w_{k}^{i}$ in $W^{i}=\left\{w_{k}^{i}\in R^{n^{i}}|k=1,\ldots \ldots ,M^{i}\right\}$, which has the same dimension as the feature vectors of $F_{i}$. The prototype vector elements are initiated with random values or to the mean of the feature vectors in $F_{i}$. During training, we are minimizing the GLVQ cost function as defined in (Equation 2). While minimizing the error between the predicted class and the ground truth, the GLVQ cost function has the advantage of being less sensitive to the initial values of the training prototypes compared to the origin LVQ [23].

##### Sensors Contribution

This part concerns the evaluation of the sensors inside the fusion system regarding each sensor’s importance in the classification task. The contribution score of one sensor is referred to by a parameter $\alpha_{i}$ expressed in percent. Once the primer distance training is achieved, the user obtains *S* times the number of prototypes that should present all the predefined classes, says *S*x$M^{i}$ prototypes. For the reminder, *S* is the total number of sensors considered in the sensor fusion system. A user who owns a sensor fusion system would be interested in having one multi-input classifier that would be used for classification. Therefore, the goal is to retrain $M^{i}$ prototypes based on the already trained *S*x$M^{i}$ prototypes. In this case, an additional custom layer is added on top of the previous layers where the training weights would be our evaluation parameters. The new prototypes will have the same dimension $n^{i}$ with the weighted sum *D* of all distances $d^{i}=d\left(f_{j}^{i},w^{i}\right)$ defined as follows:(4)$$D=\sum_{i=1}^{S}\alpha_{i}\cdot d^{i}$$
with non-negative weights $\alpha_{i}\geq 0$, which are initialized to the value 1. To convert the values of the weights into percentages, we add a softmax layer at the end of the full network.

#### 2.2.2. Sensors Robustness under Different Conditions

In this section, the evaluation of the sensor data is performed after the training of the classifiers. In fact, a validation set of data different from the one used for training is prepared for the evaluation procedure. For each sensor, we separately train a prototype-based classifier dedicated to solve one predefined classification problem. Once trained, the classifier is used for new data classification. The main goal of these tests consists of evaluating each trained classifier when there is noise and when there is no noise applied to the input validation data. Evaluation concerns both the predictive power (using validation data without noise) and the robustness (using noisy validation data) of each sensor. By integrating the reject option while performing the classification of the data, the evaluation procedure is easier to interpret. One sensor is more robust when the corresponding classifier trained on data coming from this sensor, correctly rejects more noisy data points for a fixed number of wrong rejects. By the same logic, one sensor has better prediction ability when its corresponding classifier correctly rejects more validation data compared to the other sensors, while the number of wrongly rejected data does not change, in case that no noise is applied to the used data for testing. In order to ensure a fair comparison of the evaluation outputs, the prototype-based classifiers for all sensors in the fusion system are initiated with the same number of prototypes per class.

##### Classification with Reject Option

The goal of solving a classification problem is to train a classifier with the best possible accuracy. Therefore, it would be interesting to track the cases when a classifier commits errors. The implementation of the reject option in the classification process is a good tool here to indicate how far a classifier could be trusted. In practice, there are two main reasons for uncertain classification, which are [24]:Ambiguity: The classification is uncertain because the data point falls in an overlapping region between classes. So, the decision is unclear and confusing for the classifier.Outliers: The data point is defined by new features that have not been seen before by the classifier. It can belong to another strange class for the classifier.

The classification with reject option is based on Chow, C. K.’s theory [25]. It has been applied to a different classification method, like the SVM-based classification [26], nearest neighbor classification [27] and others. In this paper, we focus on implementing the reject option for the prototype-based classification. Compared to the classical multilayer perceptron or the SVM models, we concentrate on dissimilarities and not on inner products. This helps the better interpretation of the network and the introduction of a threshold to define a local reject decision. The idea and implementation is based on L. Fischer et al.’s papers [16,28,29], where a deep explanation of the reject feature integration is presented. An efficient rejection strategy starts with a good choice of a certainty measurement *r* which is connected to the classifier model. As mentioned in Section 2.1, the GLVQ is based on a cost function that approximates the hypothesis margin, i.e., the classifier function is the relative value of the distances between the prototype of the correct and the prototype of another class [19,30]. This description of the margin can be directly used as a certainty measure during training, i.e., [28]
(5)$$\begin{array}{c} r\left(f_{j}\right)=\left|\frac{d\left(f_{j},w^{-}\right)-d\left(f_{j},w^{+}\right)}{d\left(f_{j},w^{-}\right)+d\left(f_{j},w^{+}\right)}\right| \end{array}$$
is taking the distance to the decision border into account. Another important parameter in the rejection strategy is a threshold θ, which can be a single value (global threshold) or a vector of values (local thresholds). In this study, we consider to work with a global thresholds for all the predefined classes. This parameter will define if a data point would be rejected or not. More precisely,
(6)$$\begin{array}{cc} a\;\mathrm{data}\;\mathrm{point}\;f_{i}\;\mathrm{is}\; & \left\{\begin{array}{cc} \;\mathrm{true}\;\mathrm{rejected} & \mathrm{if}\;-\theta <r(f_{j})<0\\ \;\mathrm{wrong}\;\mathrm{rejected} & \mathrm{if}\;0<r(f_{j})<\theta \\ \;\mathrm{not}\;\mathrm{rejected} & \mathrm{otherwise}\;(\left|r(f_{j})\right|>\theta ) \end{array}\right. \end{array}$$

In these cases, true reject means, that a data point would be wrongly classified without rejection and wrong reject means that the data point is mistakenly rejected, i.e., it would be correctly classified. A graphical presentation of the prototype-based classification with the rejection option integration is illustrated in Figure 3.

It is of course beneficial to reject a wrong classified data point rather than providing a false output. However, there is always a risk of rejecting correctly classified data points in case of a bad selection of the parameter θ.

The reject decision can be either online or offline. Online rejection is made during the classifier training, where the rejected data points, which present bad certainty values, are excluded and are no longer considered for training during the following epochs. The application of the rejection algorithm offline presents an evaluation tool, where new testing data points and a trained prototype-based classifier are considered. An interpretation of the system output gives an overview of the robustness of the considered classifier. In the following section, we focus on the offline evaluation of the sensors’ input based on an interpretable application of the rejection option. For a dynamic selection of the parameter θ and therefore a comparison of the behavior of the different classifiers against the change in the θ value, we refer to the dynamic programming (DP) algorithm [17]. After training, during the working phase, the margin has to be estimated slightly modifying (Equation 5), because the label information for best matching correct and best matching incorrect are not available. For this purpose, we denote the overall best matching prototype by $w^{*}$, whereas $w^{**}$ is the best matching prototype of the remaining prototypes but with a different class label than $w^{*}$, i.e., $y\left(w^{*}\right)\neq y\left(w^{**}\right)$. Then, the margin is calculated as
(7)$$\begin{array}{c} r\left(f_{j}\right)=\frac{d\left(f_{j},w^{**}\right)-d\left(f_{j},w^{*}\right)}{d\left(f_{j},w^{**}\right)+d\left(f_{j},w^{*}\right)} \end{array}$$
in analogy to Equation (Equation 5).

## 3. Experimental Results and Discussion

In this section, we consider, for evaluation, two different industrial fusion systems: time of flight (ToF)/radar fusion system for human detection and the audio/pressure fusion system for events classification. For classifiers training, we refer to the Tensorflow framework and the Keras libraries. For prototype-based learning, we refer to the Protoflow libraries [31].

### 3.1. Dynamic Evaluation of Fusion System

#### 3.1.1. Tof/Radar Fusion System

The fusion system is composed of a CamBoard pico Monstar ToF camera from the company PMD Technologies AG [32] and a 60 GHz Radar produced at Infineon Technologies AG [33]. The sensor data fusion purpose consists mainly of the classification of samples into images, which contain humans inside (positive), and images that contain objects other than humans (negative). Since we receive two different kinds of image information from the ToF camera, amplitude and depth, we assume that we possess 2 different sensors. From the radar sensor, we apply signal preprocessing algorithms on the raw data [34] to get to the end of an aligned radar image. In this test, since we focus on reducing the engineering effort, it is sufficient to consider small data sets. For each of the three sensors, we create a data set composed of 100 positive images and 60 negative images. the samples are fed in parallel into the feature extraction network. In this case, we consider a pre-trained Resnet50 neural network which is trained on the COCO dataset that include the class “Person”. Therefore, we choose to use this feature extraction network and fine-tune its pre-trained weights to our relatively small dataset. The integration of the ToF/radar fusion system into our proposed approach in order to perform both functions, classification, and evaluation, is illustrated in Figure 4.

#### 3.1.2. Audio/Pressure Fusion System

The audio/pressure fusion system is designed to be used in different industrial areas, as well as for home applications. It is an alarm system that sends signals in case of an incident or unusual events. The system is composed of a XENSIV™ MEMS microphone [35] and a digital XENSIV™ barometric pressure sensor [36], both produced at Infineon technologies AG. We collect, from each of the two sensors, 100 samples for each of the four predefined classes: open door, close door, glass break, and others. The adapted architecture to our method for the audio/pressure fusion system is presented in Figure 5.

Each sensor’s raw data are preprocessed before running the feature extraction operation. For the audio signal, the mel-log frequencies are calculated out of the raw data and then used as input for the feature extraction network. This frequency wrapping presents a better representation of the original inputs. For the pressure signal, a peak detector is applied by comparing the current sample value with the average values of the previous samples. One sample is considered in the feature extraction network only if the difference is bigger than a predefined threshold. The considered feature extraction neural network are trained from scratch using 144,110 training samples for each sensor. As illustrated in Figure 5, one convolution block is composed of three successive one-dimensional convolutional layers with reference to rectified linear unit (ReLU) as activation function. A deeper description of the systems, sensors configuration, and visualizations of the data sets with dimensions reduction from both fusion systems can be found with details in our related paper [15].

#### 3.1.3. Sensors Contribution Evaluation

We run the training for both fusion systems for 100 epochs.Since we are using small data sets and that one of the main goals of this approach is to evaluate sensors with minimum effort regarding data collection we tried different small batch sizes and figured out that the best choice is considering a batch size equal to 7. With this value, together with a learning rate equal to $10^{-4}$ for the ToF/radar fusion system and equal to $10^{-3}$ for the audio/pressure fusion system, the training models converge to the highest training accuracies and to the lowest training losses. In the end, we obtain two multi-input models each corresponding to one of the considered fusion systems. We evaluate our model using a small test set. For the ToF/radar classification model, we have an accuracy of 90.2%, whereas for the audio/pressure classifier, we have only 50% accuracy. Besides, we track the evolution of weights of the AlphaDistance layer, which present our evaluation parameters for each sensor contribution in the fusion system. Figure 6 and Figure 7 present plots of the weights’ values $\alpha_{k}$ through the training epochs for each of the two considered fusion systems.

From the Figurse Figure 6 and Figure 7, we note the two following assumptions:For the ToF/radar fusion system, amplitude information is more contributing to the classification task with 46%, then the depth information with 30% and finally in the last place the radar information with 22%.For the audio/pressure fusion system, the audio features are slightly more contributing to the events’ classification, being around 7% more than the pressure features.

In order to prove the above assumptions and therefore validate the efficiency of our evaluation approach, we run separate training using the independent data sets of the two fusion systems. Final training metrics for different prototype initialization are presented in Table 1 and Table 2. For each classifier type, the first raw presents the training accuracies and the second raw presents the training losses.

### 3.2. Static Evaluation of Fusion System

This part concerns the evaluation of both ToF and radar sensor data when there is no noise applied to the data as well as when we apply artificial noise to the validation data. We consider, in this part, classifiers trained on each sensor data set with one prototype for the positive class (yes person) and two prototypes for the negative class (no person). The visualization of the training data with corresponding trained prototypes for the three data sets is presented in Figure 8. A dimension reduction to the 2D space using the t-SNE transformation is also considered here.

After applying the DP algorithm using each classifier and validation set, we track the classifier behavior for different values of the parameter θ, which will define the reject decision border. We choose to work with a range of values of θ defined as follows:(8)$$\begin{array}{cc} \theta =[0,..,0.8] & \;\mathrm{with}\;\mathrm{an}\;\mathrm{incremental}\;\mathrm{step}\;\mathrm{of}\;0.01 \end{array}$$

In order to interpret the classifiers behaviors, we consider two kinds of plots which are:The number of wrongly rejected data points $E_{\theta }$ depending on the number of correctly rejected data points $L_{\theta }$.The accuracy reject curve (ARC), which presents the accuracy of the classifier depending on the ratio of remaining data points from the initial set after each rejection step.

#### 3.2.1. Predictive Power

This evaluation part concerns comparing the power of each trained classifier corresponding to each sensor in predicting classes with minimum classification errors. We consider a validation set composed of 100 samples from the positive class and 60 samples from the negative class. On Figure 9, we plot three curves corresponding to $E_{\theta }$ regarding $L_{\theta }$ when applying the classification with the reject option for different values of the parameter θ.

For a fixed number of wrong reject data points e.g., $E_{\theta }$ = 20, amplitude classifier is able to correctly reject less data points ca. $L_{\theta }$ = 4, then radar with $L_{\theta }$ = 5 and finally depth with $L_{\theta }$ = 6. for small values of correctly rejected data points, the values of the wrongly rejected data points are similar for all three classifiers. To get more correctly rejected data points, the number of the wrongly rejected data points is rising faster for the amplitude classifier until saturation for a value $L_{\theta }$ = 6. For both depth and radar classifiers, it is possible to have more true rejected data points with more wrong rejected data points for the θ ranges values that we choose. Another way to evaluate the rejection strategy and therefore the classifiers behavior is to track the accuracy values of the classifier on the updated validation data set with remaining points after performing the rejection task. In Figure 10, we plot three curves corresponding to the three classifiers, presenting each accuracy of the classifier while considering $X_{\theta }$, which is the remaining data points out of *X* after applying the reject option for different values of θ. $X_{\theta }$ is defined as follows:(9)$$X_{\theta }=L_{\theta }\cup E_{\theta }$$

From Figure 10, we note that the amplitude classifier has a very good accuracy independent from the ratio $X_{\theta }$/*X*, whereas for both depth and radar classifiers, the rejection feature integrated in the classification procedure affects negatively the classifier performance in making classification decisions. To conclude, the evaluation of the sensors in the fusion system regarding their prediction powers by interpreting the corresponding classifiers’ behavior while classifying new data points, reveals that amplitude classifier has better ability in making classification decisions compared to both depth and radar classifiers. In fact, depth and radar classifiers present confusions in predicting right classes when there are more rejected data points. Besides, this test allows us to confirm that looking only in the relation between $L_{\theta }$ and $E_{\theta }$ is insufficient to compare different sensors performance. We need in addition to track the corresponding classifiers accuracies for different rejection scenarios.

#### 3.2.2. Robustness

In this part of the study, we consider the same validation set as in Section 3.2.1 and we apply artificial noise. Four kinds of artificial noises are considered, which are:Gaussian noise or white noiseSpeckle noisePoisson noise or photon shot noiseSalt and Pepper noise

Gaussian and Speckle noises are independent noises, since they are characterized by a parametric distribution function that defines the noise intensity and which will be added to the original image, whereas Poisson and Salt and Pepper are dependent noises because their application is performed directly on the original image. Poisson noise occurs when two adjacent pixels inside the image are supposed to have a similar value but they end up with different intensity values. Salt and Pepper noise presents an error in data transmission so that information is lost in some of the image pixels (designed by either black or white pixels). In general, the considered noises are physical noises, linked to physics constraints like the corpuscular nature of light [37]. Since in the ToF/Radar fusion system, samples are images captured under different exposure light conditions and that sensor temperature can rise considerably for long processing hours, choosing the above-selected noise models would create evaluation scenarios similar to real-world cases. we refer to particular functions from the python numpy library in order to generate the artificial noises cited above.

Figure 11 presents one randomly selected sample from the positive class in each data set, on which we apply the different artificial noises one by one. We plot as well the histogram of the discrete cosine transform (DCT) coefficient [38]. This compressed version of the sample gives a clearer idea about the intensity of the applied noise and how much does it affect one sample.

In the same way as in Section 3.2.1, we base our evaluation on two types of curves, which are the $E_{\theta }$ regarding $L_{\theta }$ and the ARC. Each test consists of the following steps:Consider the same testing sets as in Section 3.2.1 for the three sensor types.Apply one by one the cited above noises to all the samples and store the result samples into new data setsUse the same trained model for each sensor from Section 3.2 to predict the class of the test samples with consideration of the reject option.Track for different θ values the models certainties when considering the reject option in the classification process.from Figure 12, Figure 13, Figure 14 and Figure 15, we present the output curves, which show the behavior of the 3 classifiers in the presence of different noises. Our evaluation procedure in this part is based on a comparison to the curves we obtain in Section 3.2.1 when there is no noise applied, which serve here as reference curves.

From the figures, we conclude that the three models perform in different ways in presence of noisy data:Amplitude model is not stable against Gaussian, Poisson, and Salt and Pepper noises especially when there are no more possible correctly rejected data points (in case of saturation), whereas, it presents good stability against the Speckle noise.Depth is not stable to Salt and Pepper noise, but relatively stable to Gaussian, Poisson, and Speckle noisesRadar is stable to all of the considered noises.

## 4. Conclusions

In this study, a bi-functional system for classification and evaluation purposes has been proposed. The approach is applied to sensor fusion systems aiming to solve classification problems. The proposed approach has the advantage to be independent of the sensors’ inputs and from the application. Another big advantage is that it preserves the engineering effort and time. Considering one fusion system and feeding its inputs to our adaptive proposed approach gives the user the ability to decide on the right sensor combination to consider and how far a sensor can be trusted under different conditions. From the evaluation, we focus on 3 different features. The first is the importance of each sensor in the fusion system in classifying events with high accuracy. The second feature in the evaluation tool tests the ability of each sensor in classifying events with minimum classification errors. And finally, the evaluation concerns the robustness of each sensor against noise and uncertain events. The proposed approach is applied to two different industrial fusion systems for test and validation.

Furthermore, we showed how to benefit in complex sensor systems from interpretable AI models for intelligent sensor fusion. In particular, we demonstrated that sensor weighting improves the system abilities and allows trustworthiness regarding model decision by means of intelligent reject strategies.

As a next step, on one hand, further evaluation improvements would be considered. A purely online evaluation would be interesting by extending the system architecture for training extra parameters that evaluate the sensors’ performance. This would result in large benefits in terms of engineering effort, time, and energy consumption. Moreover, this helps in generalizing the application of our proposed approach to more complex fusion systems, e.g., in autonomous driving where high precision and safety are of high importance. On the other hand, we would like to focus on the selection of features which are the basic inputs that categorize our approach, by comparing different feature extraction methods for each of the considered study cases. Furthermore, we would like to do more research on artificial noises and try to generate more realistic noises using e.g., generative adversarial networks, which can be applicable to different sensors category and at the same time correlate with noises that may occur in the real world.

## Figures and Tables

**Figure 1 sensors-21-04405-f001:**
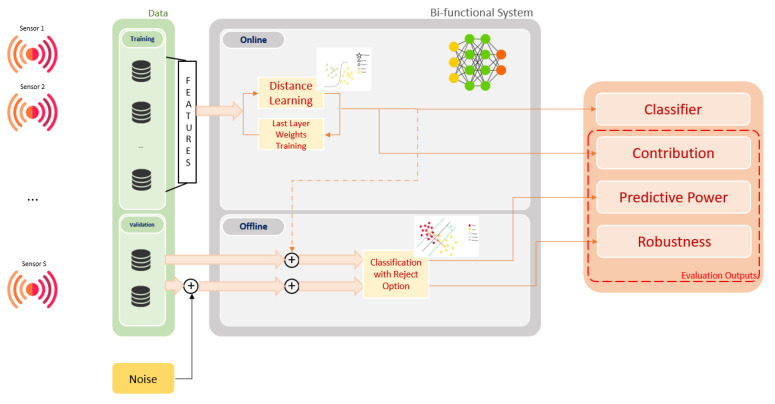
Bi-functionality of the proposed approach: Classification and Validation.

**Figure 2 sensors-21-04405-f002:**
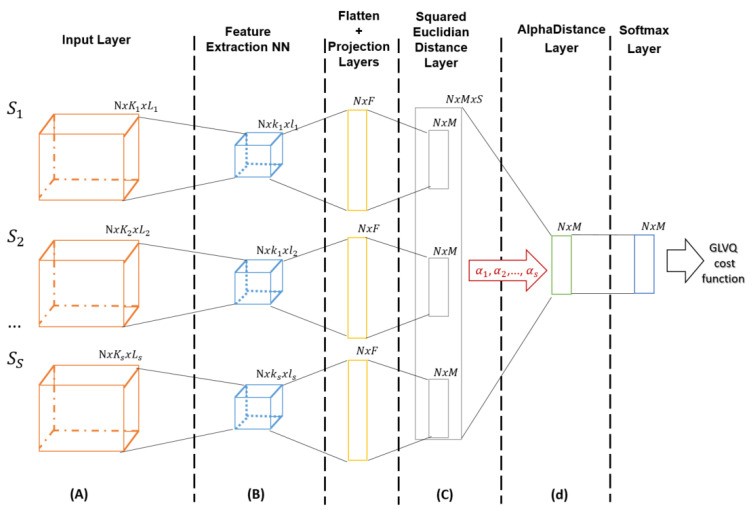
Multi-input bi-functional system layers with corresponding dimensions specification.

**Figure 3 sensors-21-04405-f003:**
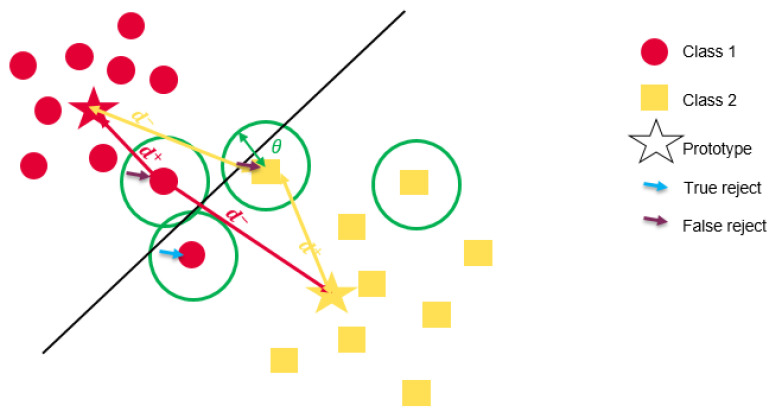
Prototype-based classification with reject criteria for a two classes problem. Each data point has a local reject decision depending on θ illustrated by the circle.

**Figure 4 sensors-21-04405-f004:**
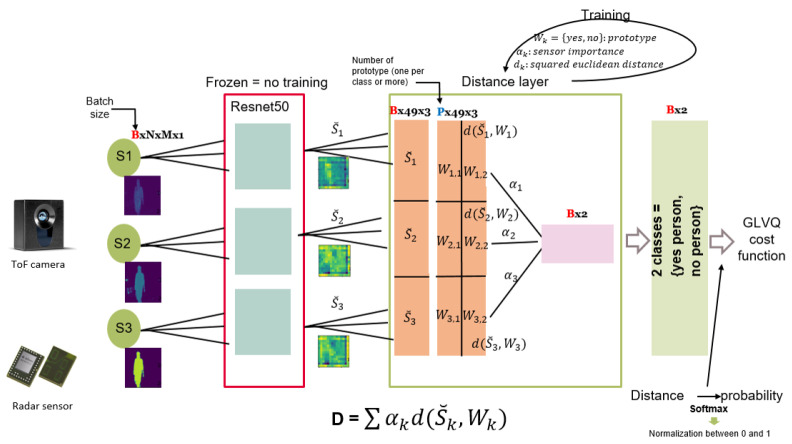
ToF/Radar fusion system for image (human) classification and sensors’ contribution evaluation.

**Figure 5 sensors-21-04405-f005:**
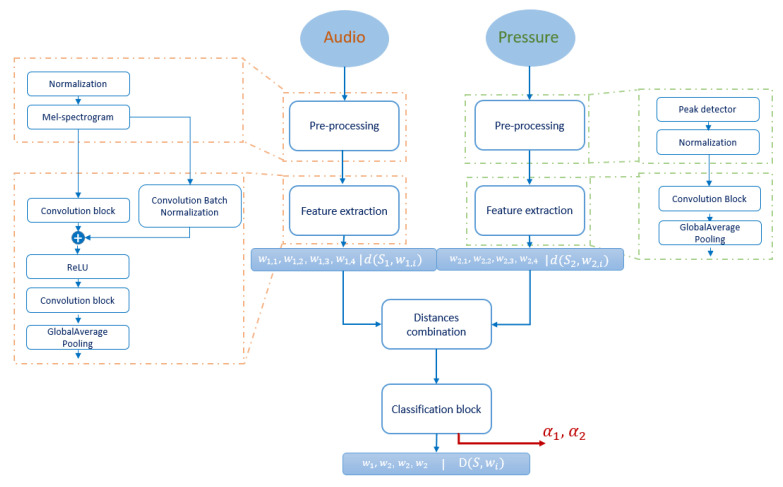
Audio/pressure fusion system for four events classification (open door, close door, glass break, other events) based on prototype learning procedure, and for sensors’ contribution evaluation.

**Figure 6 sensors-21-04405-f006:**
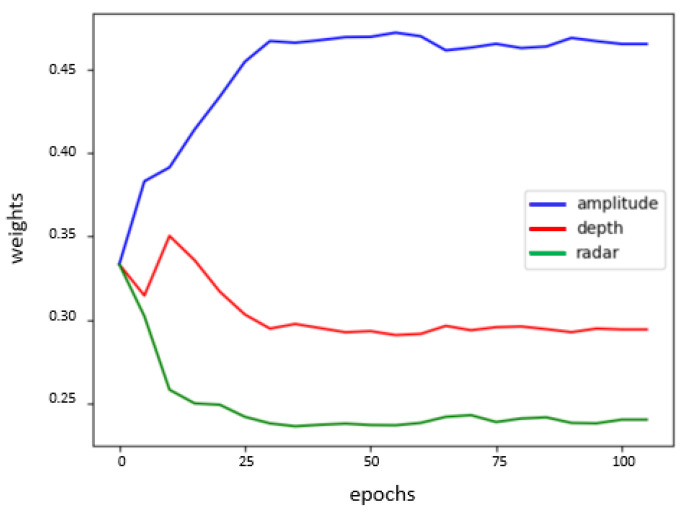
Evolution of the contribution parameters over 100 training epochs for the ToF/radar fusion system.

**Figure 7 sensors-21-04405-f007:**
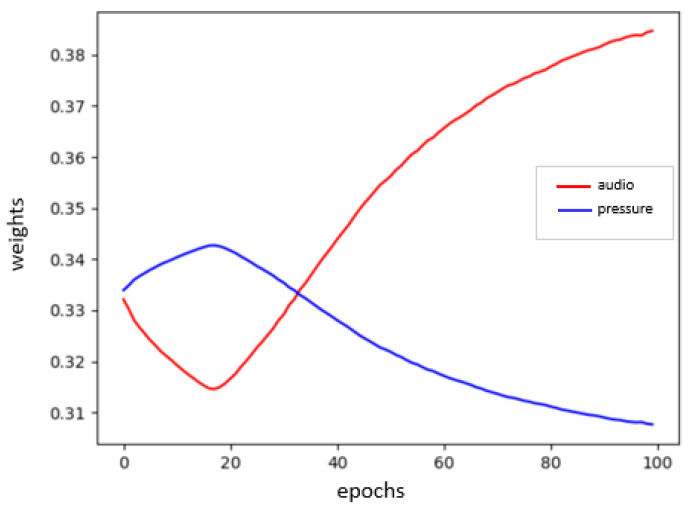
Evolution of the contribution parameters over 100 training epochs for the audio/pressure fusion system.

**Figure 8 sensors-21-04405-f008:**
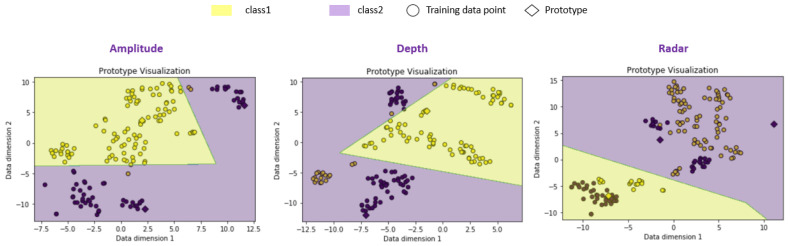
Data points and prototypes visualization in the two-dimensional space visualization after training for the three considered data sets: amplitude, depth and radar. The colored areas mark the class decision areas determined by the model.

**Figure 9 sensors-21-04405-f009:**
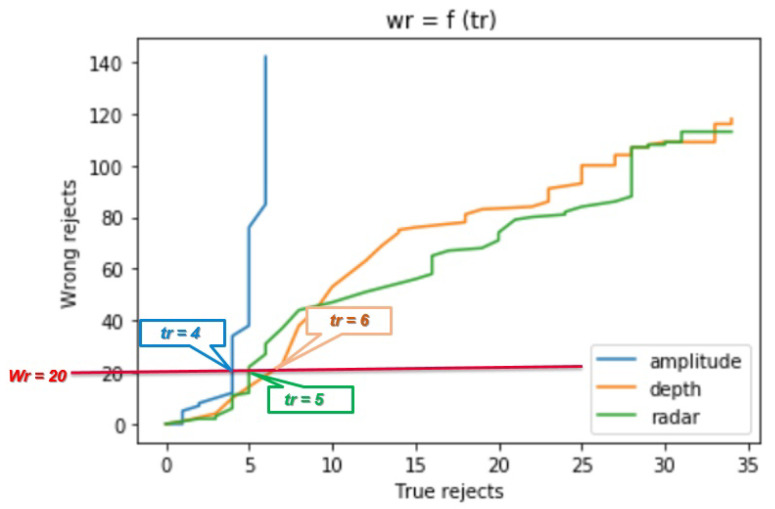
Number of wrongly rejected data points regarding the number of correctly rejected data points for the three classifiers corresponding to the three considered sensors.

**Figure 10 sensors-21-04405-f010:**
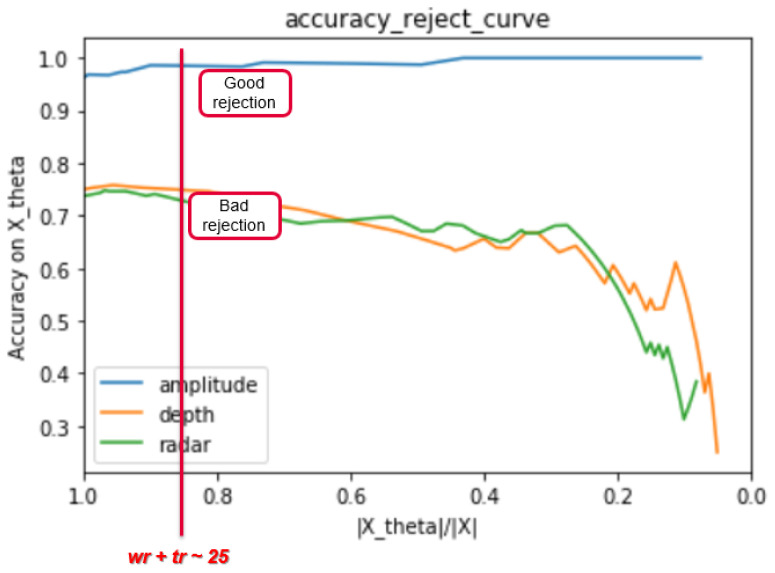
Accuracy reject curves presenting the accuracy of the three classifiers corresponding to the three considered sensors on the remaining data pints inside the validation data set after performing the classification with reject option algorithm.

**Figure 11 sensors-21-04405-f011:**
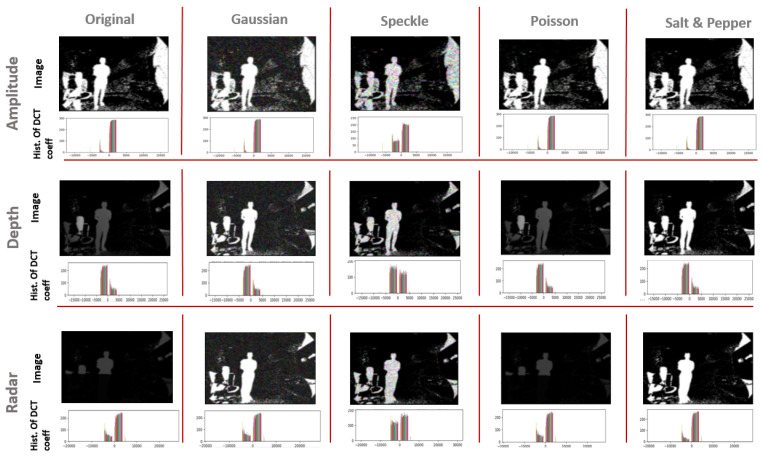
Artificial noise applied to the three different samples amplitude, depth and radar, and their corresponding DCT coefficient.

**Figure 12 sensors-21-04405-f012:**
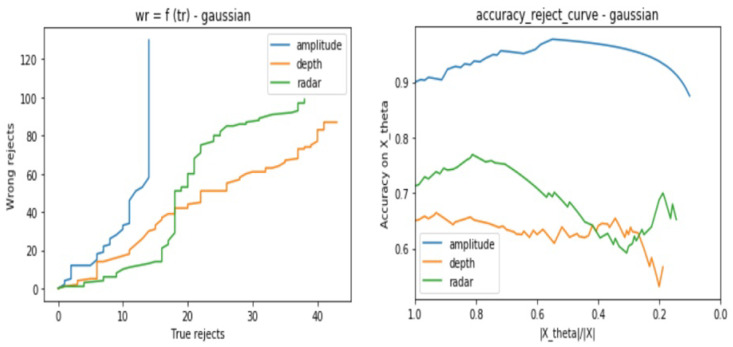
Three classifiers behavior evaluation curves when performing classification with reject option on noisy data sets—considering Gaussian noise.

**Figure 13 sensors-21-04405-f013:**
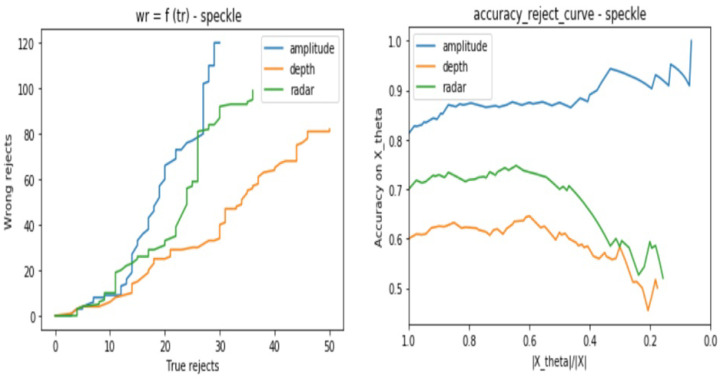
Three classifiers behavior evaluation curves when performing classification with reject option on noisy data sets—considering Speckle noise.

**Figure 14 sensors-21-04405-f014:**
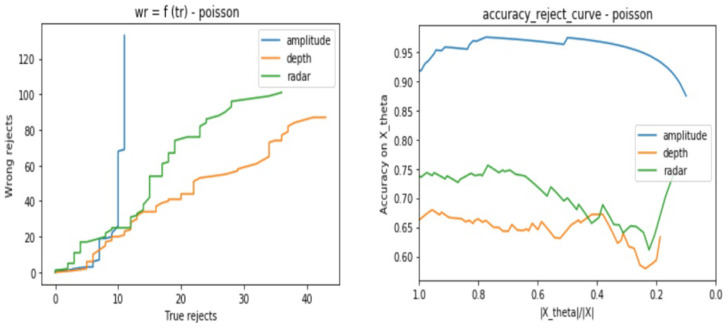
Three classifiers behavior evaluation curves when performing classification with reject option on noisy data sets—considering Poisson noise.

**Figure 15 sensors-21-04405-f015:**
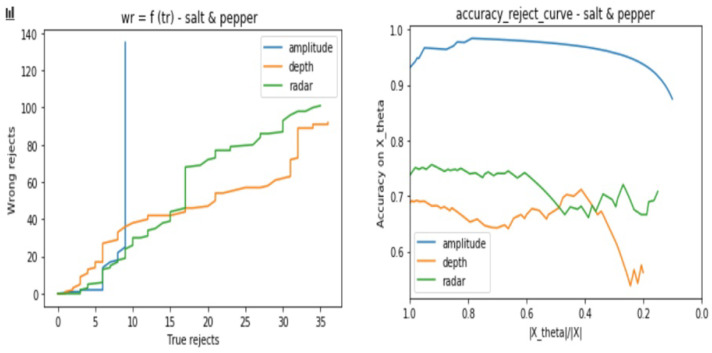
Three classifiers behavior evaluation curves when performing classification with reject option on noisy data sets—considering Salt and Pepper noise.

**Table 1 sensors-21-04405-t001:** Single-input classifiers training using each of the three data sets corresponding to sensors inside the ToF/Radar fusion system.

Classifier	1 Prototype per Class	2 Prototypes per Class	3 Prototypes per Class
Amplitude	95%	95%	95%
	0.0890	0.0843	0.0838
Depth	83.75%	84.38%	83.75%
	0.2022	0.1839	0.1856
Radar	75.63%	76.25%	76.25%
	0.2558	0.2431	0.4115

**Table 2 sensors-21-04405-t002:** Single-input classifiers training using each of the two data sets corresponding to sensors inside the audio/pressure fusion system.

Classifier	1 Prototype per Class	2 Prototypes per Class	3 Prototypes per Class
Audio	999.5%	95.5%	66.7%
	0.0074	0.0390	0.3276
Pressure	71.25%	75%	58.25%
	0.3082	0.2611	0.4115

## Data Availability

Datasets used in this study are available under this link: https://drive.google.com/file/d/11imeOzmHdEfAe0R413bluzxa8cPV7eU2/view?usp=sharing (accessed on 17 June 2021).

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
