# Peer review of "AI-Based Multi Sensor Fusion for Smart Decision Making: A Bi-Functional System for Single Sensor Evaluation in a Classification Task"

_sensors, 2021, doi:10.3390/s21134405_

Round 1

Reviewer 1 Report

The manuscript presents a methodology for using fairly well developed tools for sensor fusion and use in decision making. 

The Introduction reviews the state of the art fairly well, but leaves it somewhat (although not entirely) unclear what exactly is the void that the manuscript fills.  (A better explanation is given in the Conclusions section.)  The sentence "In this paper, we propose a smart and effective evaluation approach dedicated
to sensor fusion applications" could be followed by what this accomplishes.

The Methodology section contains a lot of useful material, but it is unclear what is standard theory (e.g. Figure 3 is standard in SVM;  if different, a very brief note would suffice) and what is new (and possibly covered in citations).

Author Response

We would like first to thank you for giving us the opportunity to submit a revised draft of our manuscript titled “AI-based Multi Sensor Fusion for Smart Decision Making: A Bi-Functional System for Single Sensor Evaluation in a Classification Task” to”MDPI Journal, section Intelligent Sensors, special issue Developing New Methods of Computational Intelligence and Data Mining in Smart Sensors Environment”. We appreciate the time and effort that you have dedicated to providing your valuable feedback on our manuscript. We have highlighted the changes within the manuscript and here is a point-by-point response to your comments and concerns.

Point 1: The Introduction reviews the state of the art fairly well, but leaves it somewhat (although not entirely) unclear what exactly is the void that the manuscript fills.  (A better explanation is given in the Conclusions section.)  The sentence "In this paper, we propose a smart and effective evaluation approach dedicated to sensor fusion applications" could be followed by what this accomplishes.

Response 1: Thank you for your nice feedback here. You are right maybe it is insufficient to only say that “we propose a smart and effective evaluation approach dedicated to sensor fusion applications”. The sentence looks general and unclear. Therefore, we explain more about our contribution in this journal paper and what is new in our approach (Please refer to the Introduction section, lines 65-72). In fact, instead of a post evaluation of the fusion approach, we propose a method for pre-evaluation of the sensors’ signals. This approach will help engineers to decide on the combination of the sensors and create values out of the fused information with less effort and within a small development time

Point 2: The Methodology section contains a lot of useful material, but it is unclear what is standard theory (e.g. Figure 3 is standard in SVM; if different, a very brief note would suffice) and what is new (and possibly covered in citations).

Response 2: Thank you for pointing out this inaccuracy to us. The width of the reject decision is depending on the value, we corrected Figure 3 accordingly.  By the way the used hypothesis margin is a lower bound for the sample margin used in SVM (Please refer to the literature in references [30] and [38] in the paper).

Reviewer 2 Report

This paper refers to an interesting problem involving a bi-functional system for classification. The system is independent of the sensors’inputs, and it has  the ability to decide on the right sensor combination. This is a topic of great interesting and with some innovation. The paper is interesting, relevant and well written.

(1)Many hyperparameters have to be tuned to have a robust convolutional neural network, and one  important hyperparameters is the batch size. In this study, the author chose a batch size of 7. That is quite small. Is there any reason behind that? 

(2) In the Table 1,  00.2431 is incorrect.

(3) Several different kinds of noise has been added to the signal from sensor and the impact on classification algorithm has been analyzed. But not all those noises exist in different sensors. The sensor may be more sensitive to one or more kinds of noise.  Only apply those kinds of noise that exist in real situation may be more meaningful.

Author Response

We would like first to thank you for giving us the opportunity to submit a revised draft of our manuscript titled “AI-based Multi Sensor Fusion for Smart Decision Making: A Bi-Functional System for Single Sensor Evaluation in a Classification Task” to”MDPI Journal, section Intelligent Sensors, special issue Developing New Methods of Computational Intelligence and Data Mining in Smart Sensors Environment”. We appreciate the time and effort that you have dedicated to providing your valuable feedback on our manuscript. We have highlighted the changes within the manuscript and here is a point-by-point response to your comments and concerns.

Point 1: Many hyperparameters have to be tuned to have a robust convolutional neural network, and one  important hyperparameters is the batch size. In this study, the author chose a batch size of 7. That is quite small. Is there any reason behind that?

Response 1: Thank you for mentioned this point. You are right 7 batch size is a small value especially it is proved that using large batch sizes allows computational speedups during the training phase. However, we are using small datasets (160x3 for the ToF/Radar fusion system & 400x2 for the Audio/Pressure fusion system) for the validation of our approach efficiency, without missing that one of our main goals is to reduce engineering efforts in terms of large data acquisition for sensors evaluation. Therefore, we tried different small values for the batch size and found out that considering a batch size equal to 7 is the best choice, which leads the training model to converge to the highest training accuracy and the lowest training loss after a certain number of training epochs (please refer to the section 3.1.3, lines 314-320)

Point 2: In the Table 1,  00.2431 is incorrect.

Response 2: Thank you for mentioning this typo mistake. It is now corrected on the manuscript (please refer to table.1 line 6 column 3).

Point 3: Several different kinds of noise has been added to the signal from sensor and the impact on classification algorithm has been analysed. But not all those noises exist in different sensors. The sensor may be more sensitive to one or more kinds of noise. Only apply those kinds of noise that exist in real situation may be more meaningful.

Response 3: Thank you for the question. We would try here to clarify our point of view in this context. Actually, by applying these kinds of noises, we aim to reproduce physical-based noises which may occur in real life for vision sensors. This is because we are validating the robustness approach using the ToF/radar fusion system which is based on input images for classification purposes. The application of such noises does not of course make sense for any fusion system or even is the same fusion system if the received signals are different in terms of the provided information. However, for a fair comparison, we take care of applying the same noises to all sensors' inputs in order to test the approach in the same shared environment under the same conditions.

In the next steps, we would like to work more on generating realistic artificial noises or even establishing favorable conditions for sensors to generate noisy data for creating real testing scenarios (which remains a relatively complicated task to do..). Please refer to the Conclusion section 4, lines 446-448. One idea would be to use Generative Adversarial Network to learn more realistic noise.

Reviewer 3 Report

The authors present a methodology to measure the importance that the information provided by several sensors has to perform the most accurate classification possible for industrial applications.

In this sense, it is described all the relevant background to understand the issue intended to be solved: yet, there are some issues

  1. What is the reason to use a convolutional layer? Can other feature extraction algorithms be used?
  2. What are the features extracted for each study case?
  3. What is the loss function used for each study case?
  4. What is the main reason to use the pre-trained Resnet50 neural network?
  5. The basic information of the system described in section 3.1.2 must be included since the paper is not available to the readers.
    1.  
  6. Since the convolutional block described in figure 5, for the audio sensor, implies that images are used, how are they obtained?
  7. Describe the convolution operation and filter used for the pressure sensor.

Author Response

We would like first to thank you for giving us the opportunity to submit a revised draft of our manuscript titled “AI-based Multi Sensor Fusion for Smart Decision Making: A Bi-Functional System for Single Sensor Evaluation in a Classification Task” to”MDPI Journal, section Intelligent Sensors, special issue Developing New Methods of Computational Intelligence and Data Mining in Smart Sensors Environment”. We appreciate the time and effort that you have dedicated to providing your valuable feedback on our manuscript. We have highlighted the changes within the manuscript and here is a point-by-point response to your comments and concerns.

Point 1: What is the reason to use a convolutional layer? Can other feature extraction algorithms be used?

Response 1: Thank you for asking this question, we think it is worth explaining why we referred to convolutional networks for feature extraction. From one part:

  • For the ToF/Radar fusion system, our input data are images with high dimension and since there is no available dataset for both sensors, we create our relatively small dataset (this is also one goal in our approach to reducing engineering efforts in collecting tones of data for training). Therefore we refer to transfer learning and use pre-trained networks with pre-trained weights and fine-tune them to our own dataset in order to get good results. (we added this small clarification in section 3.1.1, lines 286-288) .
  • For the Audio/pressure fusion system, the convolutional neural network is trained from scratch using a dataset composed of 144110 samples (In the journal paper we consider only a dataset of 400x2 for the evaluation of the sensors). The training starts by considering a deep convolutional network and then narrow it down by eliminating each time convolutional blocks until we get a relatively small neural network which gives a good test accuracy (please refer to section 3.1.2, lines 306-309, where we added more details about the network architecture).

From the other part, it would be also interesting to compare different feature extraction approaches (like trying manual feature extraction based for example on SURF or SIFT), but this is not the focus of this journal paper since we take only care of evaluating the features that we already extract from different sensors based on the usage of CNNs, which we believe are more flexible than manual feature extraction methods.

Point 2: What are the features extracted for each study case?

Response 2: As explained above, the focus of this study is not the feature extraction part. In both sections 3.1.1 and 3.1.2, we give more details about the considered feature extraction networks’ architecture for each study case but we cannot describe the resulting features because they are automatically generated by the networks.

Point 3: What is the loss function used for each study case?

Response 3: We would like to kindly guide you to section 2.1 where we introduce the considered cost function in equation (1). This GLVQ cost function is applied to all of our trained models dedicated to the sensors’ evaluation.

Point 4: What is the main reason to use the pre-trained Resnet50 neural network?

Response 4: We refer to the pre-trained network Resnet50 for the ToF/radar fusion system which is composed mainly of 2D convolutional layers because the goal of the fusion operation is to detect a person which is also a class inside the COCO dataset used for the training of the Resnet50. Moreover, as mentioned above our dataset is small and insufficient to be used for training a neural network from scratch.

Point 5: The basic information of the system described in section 3.1.2 must be included since the paper is not available to the readers.

Response 5: Thank you for your interest in knowing more about the fusion systems we were considering in order to validate our evaluation approach. In fact, the system description, sensors calibration and synchronization, and the sensors’ signal pre-processing are not the focus of the presented journal paper. The evaluation is mainly based on the features we get from the sensors’ data after going through all the steps mentioned above. Please refer to our previous contribution “Sensors data fusion for smart decisions making A novel bi-functional system for the evaluation of sensors contribution in classification problems”, which has been accepted and presented recently in the ICIT21 conference and which will be published soon in the IEEE Xplore (date has not yet been fixed).

Point 6: Since the convolutional block described in figure 5, for the audio sensor, implies that images are used, how are they obtained?

Response 6: Thank you for mentioning this point. Actually, convolutional layers are not only applicable for images, it is also possible to use them with time series data in classification problems (Some references from the literature [1],[2]). In addition, convolutional neural networks (CNN) are highly noise-resistant and are able to extract relevant features, which are independent of time. It would be interesting to compare in this case the influence of using CNNs or different time series-dedicated networks like the famous Recurrent NNs or Long Short-Term Memories on the performance of our evaluation tool (Please refer to the Conclusion section 4, lines 445-448 where we include comparing different network architecture for feature extraction as part of our future work).

[1] Fawaz, H. I., Forestier, G., Weber, J., Idoumghar, L., & Muller, P. A. (2019). Deep learning for time series classification: a review. Data Mining and Knowledge Discovery, 33(4), 917-963. https://arxiv.org/abs/1809.04356.

[2] Gui, N., Ge, D., & Hu, Z. (2019, July). AFS: An attention-based mechanism for supervised feature selection. In Proceedings of the AAAI Conference on Artificial Intelligence (Vol. 33, No. 01, pp. 3705-3713). https://arxiv.org/abs/1902.11074

Point 7: Describe the convolution operation and filter used for the pressure sensor.

Response 7: For the pressure raw signal, a simple peak detector then a normalization of the values between -1 and 1 are applied before feeding the final signal to the feature extraction network. For the audio raw signal, mel-log frequencies are calculated and then fed to the neural network for extracting the features.

For the Audio/Pressure fusion system the preprocessing part as well as the considered convolutional blocks are explained in detail in section 3.1.2, lines 300-309.

Round 2

Reviewer 3 Report

The authors have addressed all the concerns made by this reviewer

Author Response

Thank you for your comments.